# MANAS: Multi-Agent Neural Architecture Search

## Abstract

The Neural Architecture Search (NAS) problem is typically formulated as a graph search problem where the goal is to learn the optimal operations over edges in order to maximize a graph-level global objective. Due to the large architecture parameter space, efficiency is a key bottleneck preventing NAS from its practical use. In this paper, we address the issue by framing NAS as a multi-agent problem where agents control a subset of the network and coordinate to reach optimal architectures. We provide two distinct lightweight implementations, with reduced memory requirements (1/8th of state-of-the-art), and performances above those of much more computationally expensive methods. Theoretically, we demonstrate vanishing regrets of the form $\mathcal{O}(\sqrt{T})$, with $T$ being the total number of rounds. Finally, aware that random search is an (often ignored) effective baseline we perform additional experiments on 3 alternative datasets and 2 network configurations, and achieve favorable results in comparison with this baseline and other methods.

## 1 Introduction

Determining an optimal architecture is key to accurate deep neural networks (DNNs) with good generalisation properties (Szegedy et al., 2017; Huang et al., 2017; He et al., 2016; Han et al., 2017; Conneau et al., 2017; Merity et al., 2018). Neural architecture search (NAS), which has been formulated as a graph search problem, can potentially reduce the need for application-specific expert designers allowing for a wide-adoption of sophisticated networks in various industries. Zoph and Le (2017) presented the first modern algorithm automating structure design, and showed that resulting architectures can indeed outperform human-designed state-of-the-art convolutional networks (Ko, 2019; Liu et al., 2019). However, even in the current settings where flexibility is limited by expertly-designed search spaces, NAS problems are computationally very intensive with early methods requiring hundreds or thousands of GPU-days to discover state-of-the-art architectures (Zoph and Le, 2017; Real et al., 2017; Liu et al., 2018a;b).

Researchers have used a wealth of techniques ranging from reinforcement learning, where a controller network is trained to sample promising architectures (Zoph and Le, 2017; Zoph et al., 2018; Pham et al., 2018), to evolutionary algorithms that evolve a population of networks for optimal DNN design (Real et al., 2018; Liu et al., 2018b). Alas, these approaches are inefficient and can be extremely computationally and/or memory intensive as some require all tested architectures to be trained from scratch. Weight-sharing, introduced in ENAS (Pham et al., 2018), can alleviate this problem. Even so, these techniques cannot easily scale to large datasets, e.g., ImageNet. More recently, gradient-based frameworks enabled efficient solutions by introducing a continuous relaxation of the search space. For example, DARTS (Liu et al., 2019) uses this relaxation to optimise architecture parameters using gradient descent in a bi-level optimisation problem, while SNAS (Xie et al., 2019) updates architecture parameters and network weights under one generic loss. Still, due to memory constraints the search has to be performed on 8 cells, which are then stacked 20 times for the final architecture. This solution is a coarse approximation to the original problem as show in Section 6 of this work and in Doe (2019). In fact, we show that searching directly over 20 cells leads to a reduction in test error (0.24 p.p.; 8% relative to Liu et al., 2019). ProxylessNAS (Cai et al., 2019) is one exception, as it can search for the final models directly; nonetheless they still require twice the amount of memory used by our proposed algorithm.

To enable the possibility of large-scale joint optimisation of deep architectures we contribute MANAS, the first multi-agent learning algorithm for neural architecture search. Our algorithm combines the memory and computational efficiency of multi-agent systems, which is achieved through action coordination with the theoretical rigour of online machine learning, allowing us to balance exploration versus exploitation optimally. Due to its distributed nature, MANAS enables large-scale optimisation of deeper networks while learning different operations per cell. Theoretically, we demonstrate that MANAS implicitly coordinates learners to recover vanishing regrets, guaranteeing convergence. Empirically, we show that our method achieves state-of-the-art accuracy results among methods using the same evaluation protocol but with significant reductions in memory (1/8th of Liu et al., 2019) and search time (70% of Liu et al., 2019).

The multi-agent (MA) framework is inherently scalable and allows us to tackle an optimization problem that would be extremely challenging to solve efficiently otherwise: the search space of a single cell is $8^{14}$ and there is no fast way of learning the joint distribution, as needed by a single controller. More cells to learn exacerbates the problem, and this is why MA is required, as for each agent the size of the search space is always constant.

In short, our contributions can be summarised as: (1) framing NAS as a multi-agent learning problem (MANAS) where each agent supervises a subset of the network; agents coordinate through a credit assignment technique which infers the quality of each operation in the network, without suffering from the combinatorial explosion of potential solutions. (2) Proposing two lightweight implementations of our framework that are theoretically grounded. The algorithms are computationally and memory efficient, and achieve state-of-the-art results on Cifar-10 and ImageNet when compared with competing methods. Furthermore, MANAS allows search *directly* on large datasets (e.g. ImageNet). (3) Presenting 3 news datasets for NAS evaluation to minimise algorithmic overfitting; and offering a fair comparison with a random baseline.

## 2 Related work

MANAS derives its search space from DARTS (Liu et al., 2019) and is therefore most related to other gradient-based NAS methods that use the same search space. SNAS (Xie et al., 2019) appears similar at a high level, but has important differences: 1) it uses GD to learn the architecture parameters. This requires a differentiable objective (MANAS does not) and leads to 2) having to forward all operations (see their Eqs.5,6), thus negating any memory advantages (which MANAS has), and effectively requiring repeated cells and preventing search on ImageNet. ENAS (Pham et al., 2018) is also very different: its use of RL implies dependence on past states (the previous operations in the cell). It explores not only the stochastic reward function but also the relationship between states, which is where most of the complexity lies. Furthermore, RL has to balance exploration and exploitation by relying on sub-optimal heuristics, while MANAS, due to its theoretically optimal approach from online learning, is more sample efficient. Finally, ENAS uses a single LSTM (which adds complexity and problems such as exploding/vanishing gradients) to control the entire process, and is thus following a monolithic approach. Indeed, at a high level, our multi-agent framework can be seen as a way of decomposing the monolithic controller into a set of simpler, independent sub-policies. This provides a more scalable and memory efficient approach that leads to higher accuracy, as confirmed by our experiments.

## 3 Preliminary: Neural Architecture Search

We consider the NAS problem as formalised in DARTS (Liu et al., 2019). At a higher level, the architecture is composed of a *computation cell* that is a building block to be learned and stacked in the network. The cell is represented by a directed acyclic graph with $V$ nodes and $N$ edges; edges connect all nodes $i, j$ from $i$ to $j$ where $i < j$. Each vertex $\boldsymbol{x}^{(i)}$ is a latent representation for $i \in \{1, \ldots, V\}$. Each directed edge $(i, j)$ (with $i < j$) is associated with an operation $o^{(i,j)}$ that transforms $\boldsymbol{x}^{(i)}$. Intermediate node values are computed based on all of its predecessors as $\boldsymbol{x}^{(j)} = \sum_{i<j} o^{(i,j)}(\boldsymbol{x}^{(i)})$. For each edge, an architect needs to intelligently select one operation $o^{(i,j)}$ from a finite set of $K$ operations, $\mathcal{O} = \{o_k(\cdot)\}_{k=1}^{K}$, where operations represents some function to be applied to $\boldsymbol{x}^{(i)}$ to compute $\boldsymbol{x}^{(j)}$, e.g., convolutions or pooling layers. To each $o_k^{(i,j)}(\cdot)$ is associated a

set of operational weights $w_k^{(i,j)}$ that needs to be learned (e.g. the weights of a convolution filter). Additionally, a parameter $\alpha_k^{(i,j)} \in \mathbb{R}$ characterises the importance of operation $k$ within the pool $\mathcal{O}$ for edge $(i,j)$. The sets of all the operational weights $\{w_k^{(i,j)}\}$ and architecture parameters $\{\alpha_k^{(i,j)}\}$ are denoted by $\boldsymbol{w}$ and $\boldsymbol{\alpha}$, respectively. DARTS defined the operation $\bar{o}^{(i,j)}(\boldsymbol{x})$ as

$$\bar{o}^{(i,j)}(\boldsymbol{x}) = \sum_{k=1}^{K} \frac{e^{\alpha_k^{(i,j)}}}{\sum_{k'=1}^{K} e^{\alpha_{k'}^{(i,j)}}} \cdot o_k^{(i,j)}(\boldsymbol{x}) \tag{1}$$

in which $\boldsymbol{\alpha}$ encodes the network architecture. The optimal choice of architecture is defined by

$$\boldsymbol{\alpha}^{\star} = \min_{\boldsymbol{\alpha}} \mathcal{L}^{(\mathrm{val})}(\boldsymbol{\alpha}, \boldsymbol{w}^{\star}(\boldsymbol{\alpha})) \quad \text{s.t.} \quad \boldsymbol{w}^{\star}(\boldsymbol{\alpha}) = \arg\min_{\boldsymbol{w}} \mathcal{L}^{(\mathrm{train})}(\boldsymbol{\alpha}, \boldsymbol{w}). \tag{2}$$

The final objective is to obtain a *sparse* architecture $\mathcal{Z}^{\star} = \{\mathcal{Z}^{(i,j)}\}, \forall i,j$ where $\mathcal{Z}^{(i,j)} = [z_1^{(i,j)}, \ldots, z_K^{(i,j)}]$ with $z_k^{(i,j)} = 1$ for $k$ corresponding to the best operation and 0 otherwise. That is, for each pair $(i,j)$ a *single operation* is selected.

## 4 ONLINE MULTI-AGENT LEARNING FOR AUTOML

NAS suffers from a combinatorial explosion in its search space. A recently proposed approach to tackle this problem is to approximate the discrete optimisation variables (i.e., edges in our case) with continuous counterparts and then use gradient-based optimisation methods. DARTS (Liu et al., 2019) introduced this method for NAS, though it suffers from two important drawbacks. First, the algorithm is memory and computationally intensive ($\mathcal{O}(NK)$ with $K$ being total number of operations between a pair of nodes and $N$ the number of nodes) as they require loading all operation parameters into GPU memory. As a result, DARTS only optimises over a small subset of 8 cells, which are then stacked together to form a deep network of 20. Naturally, such an approximation is bound to be sub-optimal. Second, evaluating an architecture amounts to a prediction on a validation set using the optimal set of network parameters. Learning these, unfortunately, is

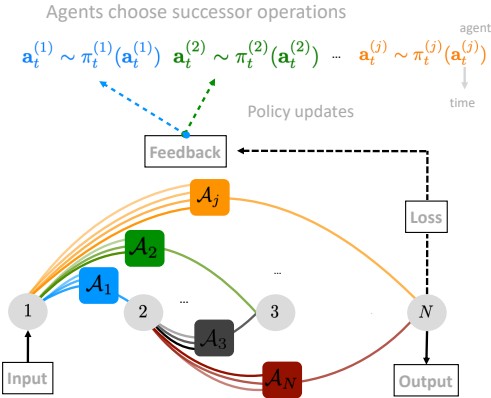

Figure 1: MANAS with single cell. Between each pair of nodes, an agent $\mathcal{A}_i$ selects action $a^{(i)}$ according to $\pi^{(i)}$. Feedback from the validation loss is used to update the policy.

highly demanding since for an architecture $\mathcal{Z}_t$, one would like to compute $\mathcal{L}_t^{(\mathrm{val})}(\mathcal{Z}_t, \boldsymbol{w}_t^{\star})$ where $\boldsymbol{w}_t^{\star} = \arg\min_{\boldsymbol{w}} \mathcal{L}_t^{(\mathrm{train})}(\boldsymbol{w}, \mathcal{Z}_t)$. DARTS, uses *weight sharing* that updates $\boldsymbol{w}_t$ once per architecture, with the hope of tracking $\boldsymbol{w}_t^{\star}$ over learning rounds. Although this technique leads to significant speed up in computation, it is not clear how this approximation affects the validation loss function.

Next, we detail a novel methodology based on a combination of multi-agent and online learning to tackle the above two problems (Figure 1). Multi-agent learning scales our algorithm, reducing memory consumption by an order of magnitude from $\mathcal{O}(NK)$ to $\mathcal{O}(N)$; and online learning enables rigorous understanding of the effect of tracking $\boldsymbol{w}_t^{\star}$ over rounds.

### 4.1 NAS AS A MULTI-AGENT PROBLEM

To address the computational complexity we use the weight sharing technique of DARTS. However, we try to handle in a more theoretically grounded way the effect of approximation of $\mathcal{L}_t^{(\mathrm{val})}(\mathcal{Z}_t, \boldsymbol{w}_t^{\star})$ by $\mathcal{L}_t^{(\mathrm{val})}(\mathcal{Z}_t, \boldsymbol{w}_t)$. Indeed, such an approximation can lead to arbitrary bad solutions due to the uncontrollable weight component. To analyse the learning problem with no stochastic assumptions on the process generating $\nu = \{\mathcal{L}_1, \ldots, \mathcal{L}_T\}$ we adopt an adversarial online learning framework.

---

**Algorithm 1** GENERAL FRAMEWORK: [steps with asterisks (*) are specified in section 5]
1: **Initialize:** $\pi_1^i$ is uniform random over all $j \in \{1, \ldots N\}$. And random $\boldsymbol{w}_1$ weights.
2: **For** $t = 1, \ldots, T$
3:     * Agent $\mathcal{A}_i$ samples $\boldsymbol{a}_t^i \sim \pi_t^i(\boldsymbol{a}_t^i)$ for all $i \in \{1, \ldots, N\}$, forming architecture $\mathcal{Z}_t$.
4:     Compute the training loss $\mathcal{L}_t^{(\text{train})}(\boldsymbol{a}_t) = \mathcal{L}_t^{(\text{train})}(\mathcal{Z}_t, \boldsymbol{w}_t)$
5:     Update $\boldsymbol{w}_{t+1}$ for all operation $\boldsymbol{a}_t^i$ in $\mathcal{Z}_t$ from $\boldsymbol{w}_t$ using back-propagation.
6:     Compute the validation loss $\mathcal{L}_t^{(\text{val})}(\boldsymbol{a}_t) = \mathcal{L}_t^{(\text{val})}(\mathcal{Z}_t, \boldsymbol{w}_{t+1})$
7:     * Update $\pi_{t+1}^i$ for all $i \in \{1, \ldots N\}$ using $\mathcal{Z}_1, \ldots, \mathcal{Z}_t$ and $\mathcal{L}_1^{(\text{val})}, \ldots, \mathcal{L}_t^{(\text{val})}$.
8: **Recommend** $\mathcal{Z}_{T+1}$, after round $T$, where $\boldsymbol{a}_{T+1}^i \sim \pi_{T+1}^i(\boldsymbol{a}_{T+1}^i)$ for all $i \in \{1, \ldots, N\}$.

---

**NAS as Multi-Agent Combinatorial Online Learning.** In Section 3, we defined a NAS problem where one out of $K$ operations needs to be recommended for each pair of nodes $(i, j)$ in a DAG. In this section, we associate *each pair* of nodes with an *agent* in charge of exploring and quantifying the quality of these $K$ operations, to ultimately recommend one. However, the only feedback for each agent is the loss that is associated with a global architecture $\mathcal{Z}$, which depends on all agents' choices.

We introduce $N$ decision makers, $\mathcal{A}_1, \ldots, \mathcal{A}_N$ (see Figure 1 and Algorithm 1). At training round $t$, each agent chooses an operation (e.g., convolution or pooling filter) according to its local action-distribution (or policy) $\boldsymbol{a}_t^j \sim \pi_t^j$, for all $j \in \{1, \ldots, N\}$ with $\boldsymbol{a}_t^j \in \{1, \ldots, K\}$. These operations have corresponding operational weights $\boldsymbol{w}_t$ that are learned in parallel. Altogether, these choices $\boldsymbol{a}_t = \boldsymbol{a}_t^1, \ldots, \boldsymbol{a}_t^N$ define a sparse graph/architecture $\mathcal{Z}_t \equiv \boldsymbol{a}_t$ for which a validation loss $\mathcal{L}_t^{(\text{val})}(\mathcal{Z}_t, \boldsymbol{w}_t)$ is computed and used by the agents to update their policies. After $T$ rounds, an architecture is recommended by sampling $\boldsymbol{a}_{T+1}^j \sim \pi_{T+1}^j$, for all $j \in \{1, \ldots, N\}$. These dynamics resemble bandit algorithms where the actions for an agent $\mathcal{A}_j$ are viewed as separate arms. This framework leaves open the design of **1)** the sampling strategy $\pi^j$ and **2)** how $\pi^j$ is updated from the observed loss.

**Minimization of worst-case regret under any loss.** The following two notions of regret motivate our proposed NAS method. Given a policy $\pi$ the *cumulative regret* $\mathcal{R}_{T,\pi}^\star$ and the *simple regret* $r_{T,\pi}^\star$ after $T$ rounds and under the worst possible environment $\nu$, are:

$$\mathcal{R}_{T,\pi}^\star = \sup_\nu \mathbb{E} \sum_{t=1}^T \mathcal{L}_t(\boldsymbol{a}_t) - \min_{\boldsymbol{a}} \sum_{t=1}^T \mathcal{L}_t(\boldsymbol{a}), \quad r_{T,\pi}^\star = \sup_\nu \mathbb{E} \sum_{t=1}^T \mathcal{L}_t(\boldsymbol{a}_{T+1}) - \min_{\boldsymbol{a}} \sum_{t=1}^T \mathcal{L}_t(\boldsymbol{a}) \quad (3)$$

where the expectation is taken over both the losses and policy distributions and $\boldsymbol{a} = \{\boldsymbol{a}^{(\mathcal{A}_j)}\}_{j=1}^N$ denotes a joint action profile. The simple regret leads to minimising the loss of the recommended architecture $\boldsymbol{a}_{T+1}$, while minimising the cumulative regret adds the extra requirement of having to sample, at any time $t$, architectures with close-to-optimal losses. We discuss in the appendix E how this requirement could improve in practice the tracking of $\boldsymbol{w}_t^\star$ by $\boldsymbol{w}_t$. We let $\mathcal{L}_t(\boldsymbol{a}_t)$ be potentially adversarilly designed to account for the difference between $\boldsymbol{w}_t^\star$ and $\boldsymbol{w}_t$ and make no assumption on its convergence. Our models and solutions in Section 5 are designed to be robust to arbitrary $\mathcal{L}_t(\boldsymbol{a}_t)$.

## 5 SOLUTION METHODS

This section elaborates our solution methods for NAS when considering adversarial losses. We propose two algorithms, MANAS and MANAS-LS, that implement two different *credit assignment techniques* specifying the update rule in line 7 of Algorithm 1. The first approximates the validation loss as a linear combination of edge weights, while the second handles non-linear loss. We propose two associated sampling techniques that specify line 3 of Algorithm 1, one minimising the *simple regret* $r_{T,\pi}^\star$ and one targeting the *cumulative regret* $\mathcal{R}_{T,\pi}^\star$, (3).

**Agent coordination, combinatorial explosion and approximate credit assignment.** Our set-up introduces multiple agents in need of coordination. Centralised critics use explicit coordination and learn the value of coordinated actions across all agents (Rashid et al., 2018), but the complexity of the problem grows exponentially with the number of possible architectures $\mathcal{Z}$, which equals $K^N$. We argue instead for an implicit approach where coordination is achieved through a joint loss function depending on the actions of all agents. This approach is scalable as each agent searches its local

action space—small and finite—for optimal action-selection rules. Both credit assignment methods below learn, for each operation $k$ belonging to an agent $\mathcal{A}_i$, a quantity $\widetilde{\boldsymbol{B}}_t^i[k]$ (similar to $\alpha$ in Section 3) that quantifies the contribution of the operation to the observed losses.

## 5.1 MANAS-LS

**Linear Decomposition of the Loss.** A simple credit assignment strategy is to approximate edge-importance (or edge-weight) by a vector $\boldsymbol{\beta}_s \in \mathbb{R}^{KN}$ representing the importance of all $K$ operations for each of the $N$ agents. $\boldsymbol{\beta}_s$ is an arbitrary, potentially adversarially-chosen vector and varies with time $s$ to account for the fact that the operational weights $\boldsymbol{w}_s$ are learned online and to avoid any restrictive assumption on their convergence. The relation between the observed loss $\mathcal{L}_s^{(\text{val})}$ and the architecture selected at each sampling stage $s$ is modeled through a linear combination of the architecture's edges (agents' actions) as

$$\mathcal{L}_s^{(\text{val})} = \boldsymbol{\beta}_s^{\mathsf{T}} \boldsymbol{Z}_s \tag{4}$$

where $\boldsymbol{Z}_s \in \{0,1\}^{KN}$ is a vectorised version of the architecture $\mathcal{Z}_s$ containing all action choices. After evaluating $S$ architectures, at round $t$ we estimate $\boldsymbol{\beta}$ by solving the following via least-squares:

$$\text{Credit assignment:} \quad \widetilde{\boldsymbol{B}}_t = \min_{\boldsymbol{\beta}} \sum_{s=1}^{S} \left( \mathcal{L}_s^{(\text{val})} - \boldsymbol{\beta}^{\mathsf{T}} \boldsymbol{Z}_s \right)^2. \tag{5}$$

Though simple, the solution gives an efficient way for agents to update their corresponding action-selection rules which they implicitly coordinate. Indeed, in Appendix C we demonstrate that the worst-case regret $\mathcal{R}_T^{\star}$ (3) can actually be decomposed into an agent-specific form $\mathcal{R}_T^i(\boldsymbol{\pi}^i, \nu^i)$ defined in the appendix: $\mathcal{R}_T^{\star} = \sup_{\nu} \mathcal{R}_T(\boldsymbol{\pi}, \nu) \iff \sup_{\nu^i} \mathcal{R}_T^i(\boldsymbol{\pi}^i, \nu^i), \quad i = 1, \dots, N$. This decomposition allows us to significantly reduce the search space and apply upcoming sampling techniques for each agent $\mathcal{A}_i$ in a completely parallel fashion.

**Zipf Sampling for $r_{T,\pi}^{\star}$.** $\mathcal{A}_i$ samples an operation $k$ proportionally to the inverse of its estimated rank $\widetilde{\langle k \rangle}_t^i$, where $\widetilde{\langle k \rangle}_t^i$ is computed by sorting the operations of agent $\mathcal{A}_i$ w.r.t $\widetilde{\boldsymbol{B}}_t^i[k]$, as

$$\text{Sampling policy:} \quad \boldsymbol{\pi}_{t+1}^i[k] = 1 \Big/ \widetilde{\langle k \rangle}_t^i \overline{\log} K \quad \text{where } \overline{\log} K = 1 + 1/2 + \dots + 1/K.$$

Zipf explores efficiently as, up to log factors, for $1 \leq m \leq K$, the $m$ estimated best operations are picked uniformly ignoring the remaining $K - m$ operations: All operations are explored almost as in uniform exploration while the estimated best is picked almost all the time. The Zipf law is anytime, parameter free, minimises optimally the simple regret in multi-armed bandits when the losses are adversarially designed and adapts optimally to stationary losses (Abbasi-Yadkori et al., 2018).

## 5.2 MANAS

**Coordinated Descent for Non-Linear Losses.** As the linear approximation is likely to be crude, an alternative is to make no assumption on the loss function and have each agent directly associate the quality of their action with the loss $\mathcal{L}_t^{(\text{val})}(\boldsymbol{a}_t)$. This results in obtaining all the agents performing a coordinated descent approach to the problem. Each agent updates for operation $k$ its $\widetilde{\boldsymbol{B}}_t^i[k]$ as

$$\text{Credit assignment:} \quad \widetilde{\boldsymbol{B}}_t^i[k] = \widetilde{\boldsymbol{B}}_{t-1}^i[k] + \mathcal{L}_t^{(\text{val})} \mathbb{1}_{\boldsymbol{a}_t^i = k} / \boldsymbol{\pi}_t^i[k]. \tag{6}$$

**Softmax Sampling for $\mathcal{R}_{T,\pi}^{\star}$.** Based on EXP3 (Auer et al., 2002), samples are from a softmax distribution (with temperature $\eta$) w.r.t. $\tilde{\boldsymbol{B}}_t^i[k]$ and the aim is to always pull the best operation as

$$\text{Sampling policy:} \quad \boldsymbol{\pi}_{t+1}^i[k] = \exp\left( \eta \tilde{\boldsymbol{B}}_t^i[k] \right) \Big/ \sum_{j=1}^{K} \exp\left( \eta \tilde{\boldsymbol{B}}_t^i[j] \right) \text{ for } k = 1, \dots, K.$$

**Comments on credit assignment.** Our MA formulation provides a gradient-free, credit assignment strategy. Gradient methods are more susceptible to bad initialisation and can get trapped in local

minima more easily than our approach, which, not only explores more widely the search space, but makes this search in an optimal way, given by the multi-armed bandit/multi-agent framework. Concretely, MANAS can easily escape from local minima as the reward is scaled by the probability of selecting an action (Eq. 6). Thus, the algorithm has a higher chance of revising its estimate of the quality of a solution based on new evidence. This is important as one-shot methods (such as MANAS and DARTS) change the network—and thus the environment—throughout the search process. Put differently, MANAS' optimal exploration-exploitation allows the algorithm to move away from 'good' solutions towards 'very good' solutions that do not live in the former's proximity; in contrast, gradient methods will tend to stay in the vicinity of a 'good' discovered solution.

## 5.3 THEORETICAL GUARANTEES

**MANAS.** This algorithms runs EXP3 (Auer et al., 2002) for each agent in parallel. If the regret of each agent is computed by considering the rest of the agent as fixed, then each agent has regret $\mathcal{O}\left(\sqrt{TK \log K}\right)$ which sums over agents to $\mathcal{O}\left(N\sqrt{TK \log K}\right)$. The proof in given in Appendix D.2.

**MANAS-LS.** We prove for this new algorithm an exponentially decreasing simple regret $r_T^\star = \mathcal{O}\left(e^{-T/H}\right)$, where $H$ is a measure of the complexity for discriminating sub-optimal solutions as $H = N(\min_{j \neq k_i^\star, 1 \leq i \leq N} \boldsymbol{B}_T^i[j] - \boldsymbol{B}_T^i[k_i^\star])$, where $k_i^\star = \min_{1 \leq j \leq K} \boldsymbol{B}_T^i[j])$ and $\boldsymbol{B}_T^i[j] = \sum_{t=1}^{T} \boldsymbol{\beta}_t^{(\mathcal{A}_i)}[j]$. The proof in given in Appendix D.1.

## 6 EXPERIMENTS RESULTS

This section, we (1) compare MANAS against existing NAS methods on the well established Cifar-10 dataset. (2) evaluate MANAS on ImageNet. (3) compare MANAS, DARTS and random sampling on 3 new datasets. Descriptions of the datasets and details of the search are provided in the Appendix. We report the performance of two algorithms, MANAS and MANAS-LS, described in Section 5.

**Search Spaces:** we use the same CNN search space as Liu et al. (2019). Since MANAS is memory efficient, it can search for the final architecture without needing to stack *a posteriori* repeated cells, and so our cells are unique. For fair comparison, we use 20 cells on Cifar-10 and 14 on ImageNet. Experiments on Sport-8, Caltech-101 and MIT-67 in Section 6.3 use both 8 and 14 cell networks. **Search Protocols:** for datasets other than ImageNet, we use 500 epochs during the search phase for architectures with 20 cells, 400 epochs for 14 cells, and 50 epochs for 8 cells. All other hyperparameters are as in Liu et al. (2019). For ImageNet, we use 14 cells and 100 epochs during search. In our experiments on the three new datasets we rerun the DARTS code to optimise an 8 cell architecture; for 14 cells we simply stacked the best cells for the appropriate number of times.

**Synthetic experiment.** To illustrate the theoretical properties of MANAS we apply it to the Gaussian Squeeze Domain experiment, a problem where agents must coordinate their actions in order to optimize a Gaussian objective function (Colby et al., 2015). MANAS progresses steadily towards zero regret while the Random Search baseline struggles to move beyond the initial starting point. Details and results are provided in Appendix F.

## 6.1 RESULTS ON CIFAR-10

**Evaluation.** To evaluate our NAS algorithm, we follow DARTS's protocol: we run MANAS 4 times with different random seeds and pick the best architecture based on its validation performance. We then randomly reinitialize the weights and retrain for 600 epochs. During search we use half of the training set as validation. To fairly compare with more recent methods, we also re-train the best searched architecture using AutoAugment and Extended Training (Cubuk et al., 2018).

**Results.** Both MANAS implementations perform well on this dataset (Table 1). Our algorithm is designed to perform comparably to Liu et al. (2019) but with an order of magnitude less memory. However, MANAS actually achieves higher accuracy. The reason for this is that DARTS is forced to search for an 8 cell architecture and subsequently stack the same cells 20 times; MANAS, on the other hand, can directly search on the final number of cells leading to better results. We also report our results when using only 8 cells: even though the network is much smaller, it still performs

Table 1: Comparison with state-of-the-art image classifiers on Cifar-10. The four row blocks represent: human-designed, NAS, MANAS search with DARTS training protocol and best searched MANAS retrained with extended protocol (AutoAugment + 1500 Epochs + 50 Channels).

| Architecture | Test Error (%) | Params (M) | Search Cost (GPU days) | Search Method |
|---|---|---|---|---|
| DenseNet-BC (Huang et al., 2017) | 3.46 | 25.6 | — | manual |
| NASNet-A (Zoph et al., 2018) | 2.65 | 3.3 | 1800 | RL |
| AmoebaNet-B (Real et al., 2018) | 2.55 | 2.8 | 3150 | evolution |
| PNAS (Liu et al., 2018a) | 3.41 | 3.2 | 225 | SMBO |
| ENAS (Pham et al., 2018) | 2.89 | 4.6 | 0.5 | RL |
| SNAS (Xie et al., 2019) | 2.85 | 2.8 | 1.5 | gradient |
| DARTS, 1st order (Liu et al., 2019) | 3.00 | 3.3 | $1.5^{\dagger}$ | gradient |
| DARTS, 2nd order (Liu et al., 2019) | 2.76 | 3.3 | $4^{\dagger}$ | gradient |
| Random + cutout (Liu et al., 2019) | 3.29 | 3.2 | — | — |
| MANAS (8 cells) | 3.05 | 1.6 | $0.8^{\dagger}$ | MA |
| MANAS (20 cells) | 2.63 | 3.4 | $2.8^{\dagger}$ | MA |
| MANAS–LS (20 cells) | **2.52** | 3.4 | $4^{\dagger}$ | MA |
| MANAS (20 cells) + AutoAugment | 1.97 | 3.4 | — | MA |
| MANAS–LS (20 cells) + AutoAugment | 1.85 | 3.4 | — | MA |

[†] Search cost is for 4 runs and test error is for the best result (for a fair comparison with other methods).

Table 2: Comparison with state-of-the-art image classifiers on ImageNet (mobile setting). The four row blocks represent: human-designed, NAS, MANAS search with DARTS training protocol and best searched MANAS retrained with extended protocol (AutoAugment + 600 Epochs + 60 Channels).

| Architecture | Test Error (%) | Params (M) | Search Cost (GPU days) | Search Method |
|---|---|---|---|---|
| ShuffleNet 2x (v2) (Zhang et al., 2018) | 26.3 | 5 | — | manual |
| NASNet-A (Zoph et al., 2018) | 26.0 | 5.3 | 1800 | RL |
| AmoebatNet-C (Real et al., 2018) | 24.3 | 6.4 | 3150 | evolution |
| PNAS (Liu et al., 2018a) | 25.8 | 5.1 | 225 | SMBO |
| SNAS (Xie et al., 2019) | 27.3 | 4.3 | 1.5 | gradient |
| DARTS (Liu et al., 2019) | 26.7 | 4.7 | 4 | gradient |
| Random sampling | 27.75 | 2.5 | — | — |
| MANAS (search on C10) | 26.47 | 2.6 | 2.8 | MA |
| MANAS (search on IN) | 26.15 | 2.6 | 110 | MA |
| MANAS (search on C10) + AutoAugment | 26.81 | 2.6 | — | MA |
| MANAS (search on IN) + AutoAugment | 25.26 | 2.6 | — | MA |

competitively with 1st-order 20-cell DARTS. This is explored in more depth in Section 6.3. Cai et al. (2019) is another method designed as an efficient alternative to DARTS; unfortunately the authors decided to a) use a different search space (PyramidNet backbone; Han et al., 2017) and b) offer no comparison to random sampling in the given search space. For these reasons we feel a numerical comparison to be unfair. Furthermore our algorithm uses half the GPU memory (they sample 2 paths at a time) and does not require the reward to be differentiable. Lastly, we observe similar gains when training the best MANAS/MANAS-LS architectures with an extended protocol (AutoAugment + 1500 Epochs + 50 Channels, in addition to the DARTS protocol).

## 6.2 RESULTS ON IMAGENET

**Evaluation.** To evaluate the results on ImageNet we train the final architecture for 250 epochs. We report the result of the best architecture out of 4, as chosen on the validation set for a fair comparison with competing methods. As search and augmentation are very expensive we use only MANAS and not MANAS-LS, as the former is computationally cheaper and performs slightly better on average.

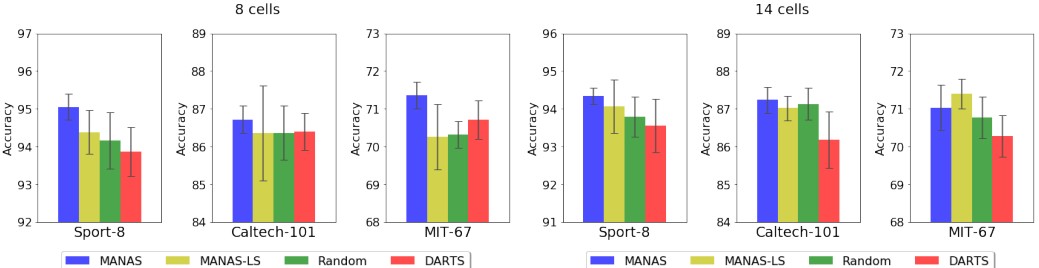

Figure 2: Comparing MANAS, random sampling and DARTS (Liu et al., 2019) on 8 and 14 cells. Average results of 8 runs. Note that DARTS was only optimised for 8 cells due to memory constraints.

**Results.** We provide results for networks searched both on Cifar-10 and directly on ImageNet, which is made possible by the computational efficiency of MANAS (Table 2). When compared to SNAS and DARTS—currently the most efficient methods, using the same search space, available—MANAS achieves state-of-the-art results both with architectures searched directly on ImageNet (0.85 p.p. improvement) and also with architectures transferred from Cifar-10 (0.55 p.p. improvement). We observe similar gains when training the best MANAS architecture with an extended training protocol (AutoAugment + 600 Epochs + 60 Channels, in addition to the DARTS protocol).

### 6.3 RESULTS ON NEW DATASETS: SPORT-8, CALTECH-101, MIT-67

**Evaluation.** The idea behind NAS is that of finding the optimal architecture, given *any* sets of data and labels. Limiting the evaluation of current methods to Cifar-10 and ImageNet could potentially lead to algorithmic overfitting. Indeed, recent results suggest that the search space was engineered in a way that makes it very hard to find a a bad architecture (Li and Talwalkar, 2019; Sciuto et al., 2019). To mitigate this, we propose testing NAS algorithms on 3 datasets (composed of regular sized images) that were never before used in this setting, but have been historically used in the CV field: Sport-8, Caltech-101 and MIT-67, described briefly in the Appendix. For these set of experiments we run the algorithm 8 times and report mean and std. We perform this both for 8 and 14 cells; we do the same with DARTS (which, due to memory constraints can only search for 8 cells). For our random baseline we sample uniformly 8 architectures from the search space. Each proposed architecture is then trained from scratch for 600 epochs as in the previous section.

**Results.** For these experiments can be found in Figure 2. MANAS manages to outperform the random baseline and significantly outperform DARTS, especially on 14 cells. It can be clearly seen from our experiments, that the optimal cell architecture for 8 cells is *not* the optimal one for 14 cells.

**Discussion on Random Search.** Clearly, in specific settings, random sampling performs very competitively. On one hand, since the search space is very large (between $8^{112}$ and $8^{280}$ architectures exist in the DARTS experiments; Liu et al., 2019), finding the global optimum is practically impossible. Why is it then that the randomly sampled architectures are able to deliver nearly state-of-the-art results? Previous experiments (Sciuto et al., 2019; Li and Talwalkar, 2019) together with the results presented here seem to indicate that the available operations and meta-structure have been carefully chosen and, as a consequence, most architectures in this space generate meaningful results. This suggests that human effort has simply transitioned from finding a good architecture to finding a good search space – a problem that needs careful consideration in future work.

## 7 CONCLUSIONS

We presented MANAS, a theoretically grounded multi-agent online learning framework for NAS. We then proposed two extremely lightweight implementations that, within the same search space, outperform state-of-the-art while reducing memory consumption by an order of magnitude compared to Liu et al. (2019). We provide vanishing regret proofs for our algorithms. Furthermore, we evaluate MANAS on 3 new datasets, empirically showing its effectiveness in a variety of settings. Finally, we confirm concerns raised in recent works (Sciuto et al., 2019; Li and Talwalkar, 2019; Doe, 2019) claiming that NAS algorithms often achieve minor gains over random architectures. We however demonstrate, that MANAS still produces competitive results with limited computational budgets.

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

## A  DATASETS

**Cifar-10.**  The CIFAR-10 dataset (Krizhevsky, 2009) is a dataset of 10 classes and consists of $50,000$ training images and $10,000$ test images of size $32\times32$. We use standard data pre-processing and augmentation techniques, i.e. subtracting the channel mean and dividing the channel standard deviation; centrally padding the training images to $40\times40$ and randomly cropping them back to $32\times32$; and randomly flipping them horizontally.

**ImageNet.**  The ImageNet dataset (Deng et al., 2009) is a dataset of 1000 classes and consists of $1,281,167$ training images and $50,000$ test images of different sizes. We use standard data pre-processing and augmentation techniques, i.e. subtracting the channel mean and dividing the channel standard deviation, cropping the training images to random size and aspect ratio, resizing them to $224\times224$, and randomly changing their brightness, contrast, and saturation, while resizing test images to $256\times256$ and cropping them at the center.

**Sport-8.**  This is an action recognition dataset containing 8 sport event categories and a total of 1579 images (Li and Fei-Fei, 2007). The tiny size of this dataset stresses the generalization capabilities of any NAS method applied to it.

**Caltech-101.**  This dataset contains 101 categories, each with 40 to 800 images of size roughly $300\times200$ (Fei-Fei et al., 2007).

**MIT-67.**  This is a dataset of 67 classes representing different indoor scenes and consists of $15,620$ images of different sizes (Quattoni and Torralba, 2009).

In experiments on Sport-8, Caltech-101 and MIT-67, we split each dataset into a training set containing $80\%$ of the data and a test set containing $20\%$ of the data. For each of them, we use the same data pre-processing techniques as for ImageNet.

## B  IMPLEMENTATION DETAILS

### B.1  METHODS

**MANAS.**  Our code is based on a modified variant of Liu et al. (2019). To set the temperature and gamma, we used as starting estimates the values suggested by Bubeck et al. (2012): $t = \frac{1}{\eta}$ with $\eta = 0.95\frac{\sqrt{\ln(K)}}{nK}$ ($K$ number of actions, $n$ number of architectures seen in the whole training). $\gamma = 1.05\frac{K\ln(K)}{n}$. We then tuned them to increase validation accuracy during the search.

**MANAS-LS.**  For our Least-Squares solution, we alternate between one epoch of training (in which all $\beta$ are frozen and the $\omega$ are updated) and one or more epochs in which we build the $Z$ matrix from Section 4 (in which both $\beta$ and $\omega$ are frozen). The exact number of iterations we perform in this latter step is dependant on the size of both the dataset and the searched architecture: our goal is simply to have a number of rows greater than the number of columns for $Z$. We then solve $\widetilde{B}_t = \left(ZZ^{\mathsf{T}}\right)^{\dagger} ZL$, and repeat the whole procedure until the end of training. This method requires no additional meta-parameters.

**Number of agents.**  In both MANAS variants, the number of agents is defined by the search space and thus is not tuned. Specifically, for the image datasets, there exists one agent for each pair of nodes, tasked with selecting the optimal operation. As there are 14 pairs in each cell, the total number of agents is $14 \times C$, with $C$ being the number of cells (8, 14 or 20, depending on the experiment).

### B.2  COMPUTATIONAL RESOURCES

ImageNet experiments were performed on multi-GPU machines loaded with $8\times$ Nvidia Tesla V100 16GB GPUs (used in parallel). All other experiments were performed on single-GPU machines loaded with $1\times$ GeForce GTX 1080 8GB GPU.

## C    FACTORIZING THE REGRET

**Factorizing the Regret:**    Let us firstly formulate the multi-agent combinatorial online learning in a more formal way. Recall, at each round, agent $\mathcal{A}_i$ samples an action from a fixed discrete collection $\{a_j^{(\mathcal{A}_i)}\}_{j=1}^K$. Therefore, after each agent makes a choice of its action at round $t$, the resulting network architecture $\mathcal{Z}_t$ is described by joint action profile $\vec{a}_t = \left[a_{j_1}^{(\mathcal{A}_1),[t]}, \dots, a_{j_N}^{(\mathcal{A}_N),[t]}\right]$ and thus, we will use $\mathcal{Z}_t$ and $\vec{a}_t$ interchangeably. Due to the discrete nature of the joint action space, the validation loss vector at round $t$ is given by $\vec{\mathcal{L}}_t^{(\mathrm{val})} = \left(\mathcal{L}_t^{(\mathrm{val})}\left(\mathcal{Z}_t^{(1)}\right), \dots, \mathcal{L}_t^{(\mathrm{val})}\left(\mathcal{Z}_t^{(K^N)}\right)\right)$ and for the environment one can write $\nu = \left(\vec{\mathcal{L}}_1^{(\mathrm{val})}, \dots, \vec{\mathcal{L}}_T^{(\mathrm{val})}\right)$. The interconnection between joint policy $\boldsymbol{\pi}$ and an environment $\nu$ works in a sequential manner as follows: at round $t$, the architecture $\mathcal{Z}_t \sim \boldsymbol{\pi}_t(\cdot|\mathcal{Z}_1, \mathcal{L}_1^{(\mathrm{val})}, \dots, \mathcal{Z}_{t-1}, \mathcal{L}_{t-1}^{(\mathrm{val})})$ is sampled and validation loss $\mathcal{L}_t^{(\mathrm{val})} = \mathcal{L}_t^{(\mathrm{val})}(\mathcal{Z}_t)$ is observed[1]. As we mentioned previously, assuming linear contribution of each individual actions to the validating loss, one goal is to find a policy $\boldsymbol{\pi}$ that keeps the regret:

$$\mathcal{R}_T(\boldsymbol{\pi}, \nu) = \mathbb{E}\left[\sum_{t=1}^T \boldsymbol{\beta}_t^{\mathsf{T}} \boldsymbol{Z}_t - \min_{\boldsymbol{Z} \in \mathcal{F}}\left[\sum_{t=1}^T \boldsymbol{\beta}_t^{\mathsf{T}} \boldsymbol{Z}\right]\right] \tag{7}$$

small with respect to all possible forms of environment $\nu$. We reason here with the cumulative regret the reasoning applies as well to the simple regret. Here, $\boldsymbol{\beta}_t \in \mathbb{R}_+^{KN}$ is a contribution vector of all actions and $\boldsymbol{Z}_t$ is binary representation of architecture $\mathcal{Z}_t$ and $\mathcal{F} \subset [0,1]^{KN}$ is set of all feasible architectures[2]. In other words, the quality of the policy is defined with respect to worst-case regret:

$$\mathcal{R}_T^* = \sup_{\nu} \mathcal{R}_T(\boldsymbol{\pi}, \nu) \tag{8}$$

Notice, that linear decomposition of the validation loss allows to rewrite the total regret (7) as a sum of agent-specific regret expressions $\mathcal{R}_T^{(\mathcal{A}_i)}\left(\boldsymbol{\pi}^{(\mathcal{A}_i)}, \nu^{(\mathcal{A}_i)}\right)$ for $i = 1, \dots, N$:

$$\mathcal{R}_T(\boldsymbol{\pi}, \nu) = \mathbb{E}\left[\sum_{t=1}^T \left(\sum_{i=1}^N \boldsymbol{\beta}_t^{(\mathcal{A}_i),\mathsf{T}} \boldsymbol{Z}_t^{(\mathcal{A}_i)} - \sum_{i=1}^N \min_{\boldsymbol{Z}^{(\mathcal{A}_i)} \in \mathcal{B}_{||\cdot||_0,1}^{(K)}(\mathbf{0})}\left[\sum_{t=1}^T \boldsymbol{\beta}_t^{(\mathcal{A}_i),\mathsf{T}} \boldsymbol{Z}^{(\mathcal{A}_i)}\right]\right)\right]$$

$$= \sum_{i=1}^N \mathbb{E}\left[\sum_{t=1}^T \boldsymbol{\beta}_t^{(\mathcal{A}_i),\mathsf{T}} \boldsymbol{Z}_t^{(\mathcal{A}_i)} - \min_{\boldsymbol{Z}^{(\mathcal{A}_i)} \in \mathcal{B}_{||\cdot||_0,1}^{(K)}(\mathbf{0})}\left[\sum_{t=1}^T \boldsymbol{\beta}_t^{(\mathcal{A}_i),\mathsf{T}} \boldsymbol{Z}^{(\mathcal{A}_i)}\right]\right]$$

$$= \sum_{i=1}^N \mathcal{R}_T^{(\mathcal{A}_i)}\left(\boldsymbol{\pi}^{(\mathcal{A}_i)}, \nu^{(\mathcal{A}_i)}\right)$$

where $\boldsymbol{\beta}_t = \left[\boldsymbol{\beta}_t^{\mathcal{A}_1,\mathsf{T}}, \dots, \boldsymbol{\beta}_t^{\mathcal{A}_N,\mathsf{T}}\right]^{\mathsf{T}}$ and $\boldsymbol{Z}_t = \left[\boldsymbol{Z}_t^{(\mathcal{A}_1),\mathsf{T}}, \dots, \boldsymbol{Z}_t^{(\mathcal{A}_N),\mathsf{T}}\right]^{\mathsf{T}}$, $\boldsymbol{Z} = \left[\boldsymbol{Z}^{(\mathcal{A}_1),\mathsf{T}}, \dots, \boldsymbol{Z}^{(\mathcal{A}_N),\mathsf{T}}\right]^{\mathsf{T}}$ are decomposition of the corresponding vectors on agent-specific parts, joint policy $\boldsymbol{\pi}(\cdot) = \prod_{i=1}^N \boldsymbol{\pi}^{(\mathcal{A}_i)}(\cdot)$, and joint environment $\nu = \prod_{i=1}^N \nu^{(\mathcal{A}_i)}$, and $\mathcal{B}_{||\cdot||_0,1}^{(K)}(\mathbf{0})$ is unit ball with respect to $||\cdot||_0$ norm centered at $\mathbf{0}$ in $[0,1]^K$. Moreover, the worst-case regret (8) also can be decomposed into agent-specific form:

$$\mathcal{R}_T^\star = \sup_{\nu} \mathcal{R}_T(\boldsymbol{\pi}, \nu) \iff \sup_{\nu^{(\mathcal{A}_i)}} \mathcal{R}_T^{(\mathcal{A}_i)}\left(\boldsymbol{\pi}^{(\mathcal{A}_i)}, \nu^{(\mathcal{A}_i)}\right), \quad i = 1, \dots, N.$$

This decomposition allows us to significantly reduce the search space and apply the two following algorithms for each agent $\mathcal{A}_i$ in a completely parallel fashion.

## D    THEORETICAL GUARANTEES

### D.1    MANAS-LS

First, we need to be more specific on the way to obtain the estimates $\tilde{\boldsymbol{\beta}}_t^{(\mathcal{A}_i)}[k]$.

---

[1] Please notice, the observed reward is actually a random variable

[2] We assume that architecture is feasible if and only if each agent chooses exactly one action.

In order to obtain theoretical guaranties we considered the least-square estimates as in Cesa-Bianchi and Lugosi (2012) as

$$\tilde{\boldsymbol{\beta}}_t = \mathcal{L}_t^{(\text{val})} \boldsymbol{P}^\dagger \boldsymbol{Z}_t \text{ where } \boldsymbol{P} = \mathbb{E}\left[\boldsymbol{Z}\boldsymbol{Z}^T\right] \text{ with } \boldsymbol{Z} \text{ has law } \boldsymbol{\pi}_t(\cdot) = \prod_{i=1}^{N} \boldsymbol{\pi}_t^{(\mathcal{A}_i)}(\cdot) \qquad (9)$$

Our analysis is under the assumption that each $\boldsymbol{\beta}_t \in \mathbb{R}^{KN}$ belongs to the linear space spanned by the space of sparse architecture $\boldsymbol{\mathcal{Z}}$. This is not a strong assumption as the only condition on a sparse architecture comes with the sole restriction that one operation for each agent is active.

**Theorem 1.** *Let us consider neural architecture search problem in a multi-agent combinatorial online learning form with $N$ agents such that each agent has $K$ actions. Then after $T$ rounds, MANAS-LS achieves joint policy $\{\boldsymbol{\pi}_t\}_{t=1}^T$ with expected simple regret (Equation 3) bounded by $\mathcal{O}\left(e^{-T/H}\right)$ in any adversarial environment with complexity bounded by $H = N(\min_{j \neq k_i^\star, i \in \{1,\dots,N\}} \boldsymbol{B}_T^{(\mathcal{A}_i)}[j] - \boldsymbol{B}_T^{(\mathcal{A}_i)}[k_i^\star])$, where $k_i^\star = \min_{j \in \{1,\dots,K\}} \boldsymbol{B}_T^{(\mathcal{A}_i)}[j]$.*

*Proof.* In Equation 9 we use the same constructions of estimates $\tilde{\boldsymbol{\beta}}_t$ as in ComBand. Using Corollary 14 in Cesa-Bianchi and Lugosi (2012) we then have that $\widetilde{\boldsymbol{B}}_t$ is an unbiased estimates of $\boldsymbol{B}_t$.

Given the adversary losses, the random variables $\tilde{\boldsymbol{\beta}}_t$ can be dependent of each other and $t \in [T]$ as $\pi_t$ depends on previous observations at previous rounds. Therefore, we use the Azuma inequality for martingale differences by Freedman (1975).

Without loss of generality we assume that the loss $\mathcal{L}_t^{(\text{val})}$ are bounded such that $\mathcal{L}_t^{(\text{val})} \in [0, 1]$ for all $t$. Therefore we can bound the simple regret of each agent by the probability of misidentifying of the best operation $P(k_i^\star \neq a_{T+1}^{\mathcal{A}_i})$.

We consider a fixed adversary of complexity bounded by $H$. For simplicity, and without loss of generality, we order the operations from such that $\boldsymbol{B}_T^{(\mathcal{A}_i)}[1] < \boldsymbol{B}_T^{(\mathcal{A}_i)}[2] \leq \ldots \leq \boldsymbol{B}_T^{(\mathcal{A}_i)}[K]$ for all agents.

We denote for $k > 1$, $\Delta_k = \boldsymbol{B}_T^{(\mathcal{A}_i)}[k] - \boldsymbol{B}_T^{(\mathcal{A}_i)}[k_i^\star]$ and $\Delta_1 = \Delta_2$.

We also have $\lambda_{min}$ as the smallest nonzero eigenvalue of $\boldsymbol{M}$ where $\boldsymbol{M}$ is $\boldsymbol{M} = E[\boldsymbol{Z}\boldsymbol{Z}^T]$ where $\boldsymbol{Z}$ is a random vector representing a sparse architecture distributed according to the uniform distribution.

$$
\begin{aligned}
P(k_i^\star \neq a_{T+1}^{\mathcal{A}_i}) &= P\left(\exists k \in \{1, \dots, K\} : \widetilde{\boldsymbol{B}}_T^{(\mathcal{A}_i)}[1] \geq \widetilde{\boldsymbol{B}}_T^{(\mathcal{A}_i)}[k]\right) \\
&\leq P\left(\exists k \in \{1, \dots, K\} : \boldsymbol{B}_T^{(\mathcal{A}_i)}[k] - \widetilde{\boldsymbol{B}}_T^{(\mathcal{A}_i)}[k] \geq \frac{T\Delta_k}{2} \text{ or } \widetilde{\boldsymbol{B}}_T^{(\mathcal{A}_i)}[1] - \boldsymbol{B}_T^{(\mathcal{A}_i)}[1] \geq \frac{T\Delta_1}{2}\right) \\
&\leq P\left(\widetilde{\boldsymbol{B}}_T^{(\mathcal{A}_i)}[1] - \boldsymbol{B}_T^{(\mathcal{A}_i)}[1] \geq \frac{T\Delta_1}{2}\right) + \sum_{k=2}^{K} P\left(\boldsymbol{B}_T^{(\mathcal{A}_i)}[k] - \widetilde{\boldsymbol{B}}_T^{(\mathcal{A}_i)}[k] \geq \frac{T\Delta_k}{2}\right) \\
&\overset{(\mathbf{a})}{\leq} \sum_{k=1}^{K} \exp\left(-\frac{(\Delta_k)^2 T}{2Nlog(K)/\lambda_{min}}\right) \\
&\leq K \exp\left(-\frac{(\Delta_1)^2 T}{2Nlog(K)/\lambda_{min}}\right),
\end{aligned}
$$

where **(a)** is using Azuma's inequality for martingales applied to the sum of the random variables with mean zero that are $\tilde{\boldsymbol{\beta}}_{k,t} - \boldsymbol{\beta}_{k,t}$ for which we have the following bounds on the range. The range of $\tilde{\boldsymbol{\beta}}_{k,t}$ is $[0, Nlog(K)/\lambda_{min}]$. Indeed our sampling policy is uniform with probability $1/log(K)$ therefore one can bound $\tilde{\boldsymbol{\beta}}_{k,t}$ as in (Cesa-Bianchi and Lugosi, 2012, Theorem 1) Therefore we have $|\tilde{\boldsymbol{\beta}}_{k,t} - \boldsymbol{\beta}_{k,t}| \leq Nlog(K)/\lambda_{min}$.

We recover the result with a union bound on all agents. $\qquad \square$

## D.2   MANAS

We consider a simplified notion of regret that is a regret per agent where each agent is considering the rest of the agents as part of the adversarial environment. Let us fix our new objective as to minimise

$$\sum_{i=1}^{N} \mathcal{R}_T^{\star,i}(\pi^{(\mathcal{A}_i)}) = \sum_{i=1}^{N} \sup_{\boldsymbol{a}_{-i},\nu} \mathbb{E}\left[\sum_{t=1}^{T} \mathcal{L}_t^{(\text{val})}(\boldsymbol{a}_t^{(\mathcal{A}_i)}, \boldsymbol{a}_{-i}) - \min_{\boldsymbol{a}\in\{1,...,K\}}\left[\sum_{t=1}^{T} \mathcal{L}_t^{(\text{val})}(\boldsymbol{a}, \boldsymbol{a}_{-i})\right]\right],$$

where $\boldsymbol{a}_{-i}$ is a fixed set of actions played by all agents to the exception of agent $\mathcal{A}_i$ for the $T$ rounds of the game and $\nu$ contains all the losses as $\nu = \{\mathcal{L}_t^{(\text{val})}(\boldsymbol{a})\}_{t\in\{1,...,T\},\boldsymbol{a}\in\{1,...,K^N\}}$.

We then can prove the following bound for that new notion of regret.

**Theorem 2.** *Let us consider neural architecture search problem in a multi-agent combinatorial online learning form with $N$ agents such that each agent has $K$ actions. Then after $T$ rounds, MANAS achieves joint policy $\{\pi_t\}_{t=1}^{T}$ with expected cumulative regret bounded by $\mathcal{O}\left(N\sqrt{TK\log K}\right)$.*

*Proof.* First we look at the problem for each given agent $\mathcal{A}_i$ and we define and look at

$$\mathcal{R}_T^{\star,i}(\pi^{(\mathcal{A}_i)}, \boldsymbol{a}_{-i}) = \sup_{\nu} \mathbb{E}\left[\sum_{t=1}^{T} \mathcal{L}_t^{(\text{val})}(\boldsymbol{a}_t^{(\mathcal{A}_i)}, \boldsymbol{a}_{-i}) - \min_{\boldsymbol{a}\in\{1,...,K\}}\left[\sum_{t=1}^{T} \mathcal{L}_t^{(\text{val})}(\boldsymbol{a}, \boldsymbol{a}_{-i})\right]\right],$$

We want to relate that the game that agent $i$ plays against an adversary when the actions of all the other agents are fixed to $\boldsymbol{a}_{-i}$ to the vanilla EXP3 setting. To be more precise on why this is the EXP3 setting, first we have that $\mathcal{L}_t^{(\text{val})}(\boldsymbol{a}_t)$ is a function of $\boldsymbol{a}_t$ that can take $K^N$ arbitrary values. When we fix $\boldsymbol{a}_{-i}$, $\mathcal{L}_t^{(\text{val})}(\boldsymbol{a}_t^{(\mathcal{A}_i)}, \boldsymbol{a}_{-i})$ is a function of $\boldsymbol{a}_t^{(\mathcal{A}_i)}$ that can only take $K$ arbitrary values.

One can redefine $\mathcal{L}_t^{@,(\text{val})}(\boldsymbol{a}_t^{(\mathcal{A}_i)}) = \mathcal{L}_t^{(\text{val})}(\boldsymbol{a}_t^{(\mathcal{A}_i)}, \boldsymbol{a}_{-i})$ and then the game boils down to the vanilla adversarial multi-arm bandit where each time the learner plays $\boldsymbol{a}_t^{(\mathcal{A}_i)} \in \{1, \ldots, K\}$ and observes/incur the loss $\mathcal{L}_t^{@,(\text{val})}(\boldsymbol{a}_t^{(\mathcal{A}_i)})$. Said differently this defines a game where the new $\nu'$ contains all the losses as $\nu' = \{\mathcal{L}_t^{@,(\text{val})}(\boldsymbol{a}^{(\mathcal{A}_i)})\}_{t\in\{1,...,T\},\boldsymbol{a}^{(\mathcal{A}_i)}\in\{1,...,K\}}$.

For all $\boldsymbol{a}_{-i}$

$$\mathcal{R}_T^{\star,i}(EXP3, \boldsymbol{a}_{-i}) \leq 2\sqrt{TK\log(K)}$$

Then we have

$$\mathcal{R}_T^{\star,i}(EXP3) \leq \sup_{\boldsymbol{a}_{-i}} 2\sqrt{TK\log(K)}$$
$$= 2\sqrt{TK\log(K)}$$

Then we have

$$\sum_{i=1}^{N} \mathcal{R}_T^{\star,i}(EXP3) \leq 2N\sqrt{TK\log(K)}$$

$\square$

# E   RELATION BETWEEN WEIGHT SHARING AND CUMULATIVE REGRET

Ideally we would like to obtain for any given architecture $\mathcal{Z}$ the value $\mathcal{L}_{val}(\mathcal{Z}, \boldsymbol{w}^\star(\mathcal{Z}))$. However obtaining $\boldsymbol{w}^\star(\mathcal{Z}) = \arg\min_{\boldsymbol{w}} \mathcal{L}_{train}(\boldsymbol{w}, \mathcal{Z})$ for any given fixed $\mathcal{Z}$ would already require heavy computations. In our approach the $\boldsymbol{w}_t$ that we compute and update is actually common to all $\mathcal{Z}_t$ as $\boldsymbol{w}_t$ replaces $\boldsymbol{w}^\star(\mathcal{Z}_t)$. This is a simplification that leads to learning a weight $\boldsymbol{w}_t$ that tend to minimise the loss $\mathbb{E}_{\mathcal{Z}\sim\pi_t}[\mathcal{L}_{val}(\mathcal{Z}, \boldsymbol{w}(\mathcal{Z})]$ instead of minimising $\mathcal{L}_{val}(\mathcal{Z}_t, \boldsymbol{w}(\mathcal{Z}_t))$. If $\pi_t$ is concentrated on a fixed $\mathcal{Z}$ then these two previous expressions would be close. Moreover when $\pi_t$ is concentrated on $\mathcal{Z}$ then $\boldsymbol{w}_t$ will approximate accurately $\boldsymbol{w}^\star(\mathcal{Z})$ after a few steps. Note that this gives an argument for using sampling algorithm that minimise the cumulative regret as they naturally tend to play almost all the time one specific architecture. However there is a potential pitfall of converging to a local minimal solution as $\boldsymbol{w}_t$ might not have learned well enough to compute accurately the loss of other and potentially better architectures.

## F  GAUSSIAN SQUEEZE DOMAIN EXPERIMENT

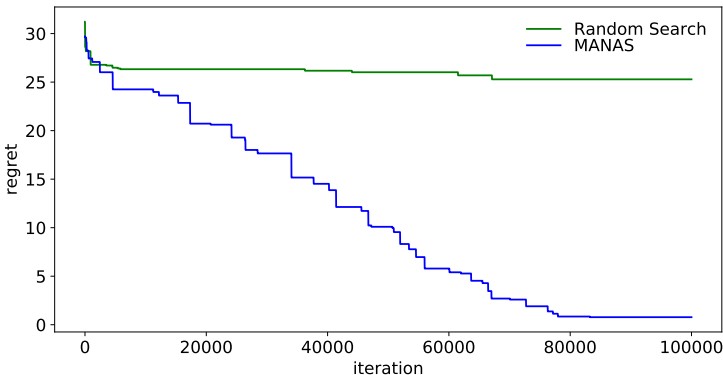

Figure 3: Regret for the Gaussian Squeeze Domain experiment with 100 agents, 10 actions, $\mu = 1$, $\sigma = 10$.

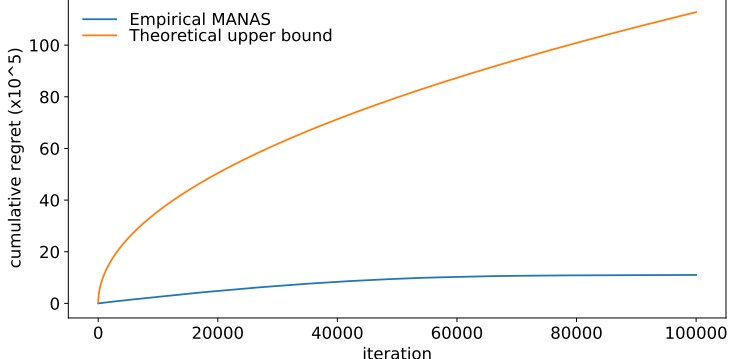

Figure 4: Theoretical bound for the MANAS cumulative regret ($2N\sqrt{TK \log K}$; see Appendix D.2) and the observed counterpart for the Gaussian Squeeze Domain experiment with 100 agents, 10 actions, $\mu = 1$, $\sigma = 10$.

