# OpenReview forum: "MANAS: Multi-Agent Neural Architecture Search"
_ICLR.cc/2020/Conference — Reject_

### Official Review · AnonReviewer1 · 2019-10-26
**Official Blind Review #1**

**Rating:** 3

**Review:**

In this paper,  the authors pay attention on the bottleneck in the NAS of its large architecture space which cause low efficiency. They introduce the multi agent reinforcement learning method to take the neural architecture search as a multi agent reinforcement learning problem.

Main contribution is :(1) Framing the MAS as a multi agent problem. (2)Purpose two lightweight implementation. (3) Presenting 3 new datasets for NAS evaluation to minimize algorithmic over-fitting.

It seems like that it is the first work to combine multi agent reinforcement learning with NAS, and you have make complete proof about the algorithm's efficiency both mathematically and empirically. But from the view of multi agent reinforcement learning, there are also some points which make me confused.

The main problem is coordination, and I understand it as the agents in your work aim to get a joint action and the training process of them are independent, but we all know that in multi agent problems, the changing of agent's policy will cause change of the environment, so it will bring the instability, so I want to know that how you deal with the instability or whether the instability influence a lot in your work? Another problem may be not a theoretically problem that I want to know that have you made the guarantee of the consistency of agents' policies when using parallel training (May be the framework in coding process guarantee it ?) or the consistency is unnecessary to talk because it doesn't influence the result?

**Experience Assessment:**

I have read many papers in this area.

**Review Assessment: Checking Correctness Of Derivations And Theory:**

I carefully checked the derivations and theory.

**Review Assessment: Checking Correctness Of Experiments:**

I carefully checked the experiments.

**Review Assessment: Thoroughness In Paper Reading:**

I read the paper at least twice and used my best judgement in assessing the paper.

---

> ### Author Response · Authors · 2019-11-11
> **Reply to Reviewer #1**
>
> We thank the reviewer for the encouraging and insightful comments. We hope the following will address the issues raised.
>
> Regarding instability:
> Our theoretical guarantees are based on the worst-case scenarios of adversarial losses, which includes potential instabilities in the joint behavior of the agents. Indeed, this can be seen as a special form of reward stochasticity, which our algorithm is robust to. In practice, we never encounter instabilities during training nor failure in convergence.
>
> Regarding, consistency of agents' policies:
> If by parallel training the reviewer means training on multiple GPUs, then this is not an issue, as we wait for the full batch step to complete on all GPU cards before computing the reward, i.e. the coordination is synchronized and therefore the agents' policies are consistent. Please let us know if we misunderstood the question.

---

### Official Review · AnonReviewer3 · 2019-10-27
**Official Blind Review #3**

**Rating:** 6

**Review:**

In this paper, the authors proposed MANAS, which is based on DARTS, by approximating the problem space by factorizing them into smaller spaces, which will be solved by multiple agents. The authors claimed that this can simplified the search space so that the joint search can be more efficient to enable us to search a larger space faster. While the overall idea seems simple but the coordinating among agents can be difficult, where the authors proposed credit assignment techniques to address the issue. The final algorithm is evaluated on CV datasets as well as 3 new datasets.

Overall, I found the motivation and proposed solution by the authors convincing. However, in a search space where random searcher is competitive, it is important for us to have an in-depth understanding on the proposed techniques. Especially when the experimental results is not fully comparable (it is difficult to control #params to evaluate the Test Error, and Search Cost being a one-time cost), I think the magnitude of the improvement showed in the experimental results itself might not be enough to justify this new approach.

I am curious about the how does the number of agents affects the experimental results. It seems that it is not mentioned in the experiment section (or I might missed it?). And do we need to search for the best number of agents, which will add to the search cost?

I am also curious on if one can combine other search algorithms with the similar idea on dividing the search space, e.g., using random search on a subspace in a coordinate descend fashion. It will be great if the authors can provide more in-depth analysis on different component of the proposed algorithm so that we can fully understand the source of improvement.


**Experience Assessment:**

I have read many papers in this area.

**Review Assessment: Checking Correctness Of Derivations And Theory:**

I did not assess the derivations or theory.

**Review Assessment: Checking Correctness Of Experiments:**

I assessed the sensibility of the experiments.

**Review Assessment: Thoroughness In Paper Reading:**

I read the paper at least twice and used my best judgement in assessing the paper.

---

> ### Author Response · Authors · 2019-11-11
> **Reply to Reviewer #3**
>
> We thank the reviewer for the very valuable comments and would like to address the issues raised.
>
> Regarding experimental results:
> The main contribution of MANAS is to lower computational costs while maintaining state-of-the-art performance. NAS is extremely expensive and, therefore, we believe that reductions in computational costs will be well received by the community and boost the adoption of this technique in real applications, where budget is typically limited.
> The reduction in search time is approximately 30% and the reduction in GPU memory consumption is almost 90% (compared to DARTS on CIFAR10 for 20-cell models). Both of these allow MANAS to tackle larger datasets and larger models, which less efficient methods could not handle.
> Given these improvements, it is remarkable that MANAS is also able to provide an increase in accuracy, even if of small magnitude (0.48/0.55 p.p. on Cifar-10/ImageNet when compared to first-order DARTS; p.p. is percentage points).
> Our experiments were designed to make comparisons with other methods as fair as possible. However, it has become clear that the training protocol is extremely important for a network's final accuracy, and current state-of-the-art methods tend to use all possible improvements (e.g. AutoAugment, extended training, more channels). To make our method more comparable with recent approaches, we also included results with a better training protocol (Tables 1 and 2). Doing this puts our method between 1.15/1.44 p.p. above DARTS on CIFAR10/ImageNet.
>
> Regarding the number of agents:
> In MANAS each agent is responsible for selecting an operation to be performed between a pair of nodes. As the number of nodes within a cell is fixed, the number of agents is dictated by the size of the network (the number of cells). Increasing the number of agents would effectively demand an increase in the number of cells, as the two are tightly coupled.
> To provide a fair comparison with previous methods, we fix the size of the network (14 or 20 cells), and so the number of agents follows directly from this choice. In the DARTS search space this translates into 14 agents per cell.
> We have clarified this in the implementation details (Appendix B.1)
>
> Regarding sources of improvement:
> This is a very important point, and we would like to clarify the benefits of our approach.
> The main source of improvement is the gradient-free, credit assignment strategy used. Gradient methods are more susceptible to bad initialisation and can get trapped in local minima more easily than our approach, which not only explores more widely the search space, but makes this search in an optimal way, given by the multi-armed bandit/multi-agent framework. Concretely, MANAS can easily escape from local minima as the reward is scaled by the probability of selecting an action (Eq.6). Thus, the algorithm has a higher chance of revising its estimate of the quality of a solution based on new evidence. This is important as one-shot methods (such as MANAS and DARTS) change the network--and thus the environment--throughout the search process. Put differently, MANAS' optimal exploration-exploitation allows the algorithm to move away from 'good' solutions towards 'very good' solutions that do not live in the former's proximity; in contrast, gradient methods will tend to stay in the vicinity of a 'good' discovered solution.
> We have clarified this in the main text (Sec.5.2).
>
> Regarding combining other search algorithms with divided search space:
> Conceptually it could be possible to combine a number of different search strategies with our intuition of locally dividing the search space. In fact, some optimisation techniques seem to follow a similar approach [1], although none has been applied to NAS.
> However, their implementation would be non-trivial in practice. For instance, random search on it's own would have no tools to coordinate between different agents and would end up following a non-optimal greedy choice for each agent. Likewise, no theoretical guarantees could be provided without further work.
>
> [1] Audet, Charles, John E. Dennis Jr, and Sébastien Le Digabel. "Parallel space decomposition of the mesh adaptive direct search algorithm." SIAM Journal on Optimization 19.3 (2008): 1150-1170.

---

### Official Review · AnonReviewer2 · 2019-10-28
**Official Blind Review #2**

**Rating:** 6

**Review:**

This work built on top of DARTS. In their setting, each edge on the DAG (same as the one proposed in DARTS) has one agent associated with it and every agent maintains weights to propose operations. The author introduced two ways to update these weights: 1) solving a least squares assuming the validation loss decomposes linearly on the operations (MANAS-LS); 2) only update the weights for the activated operations (MANAS). Due to the usage of bandit framework, theoretical guarantees on the regret can be derived.

Because the distributed nature of the agents, this work is memory efficient and it allows searching directly on large datasets. The empirical results showed competitive performance in less GPU days comparing to  DARTS and recent variants.

The paper is well written. Apart from the theoretical contributions, the empirical evaluations are well done: the author used 3 more datasets instead of the usual CIFAR-10 and IMAGENET. Also, the random search are brought into picture which I think every NAS paper should include.

It's surprising to see MANAS-LS sometimes outperform MANAS. For me, MANAS is a more principle way. Do the authors have more explanations? Why the test error of MANAS-LS + AutoAugment is missing in Table 1?

It's nice that the authors apply bandit framework to derive theoretical guarantees, but how close are these guarantees to the practice (for example on the benchmarks used in the work)? Is there some study for that? As there are not so many NAS works with theories, I think it would be nice if the authors could also comment on that.

**Experience Assessment:**

I have read many papers in this area.

**Review Assessment: Checking Correctness Of Derivations And Theory:**

I did not assess the derivations or theory.

**Review Assessment: Checking Correctness Of Experiments:**

I assessed the sensibility of the experiments.

**Review Assessment: Thoroughness In Paper Reading:**

I made a quick assessment of this paper.

---

> ### Author Response · Authors · 2019-11-11
> **Reply to Reviewer #2**
>
> We thank the reviewer for the positive feedback. We would like to address the two main concerns.
>
> Regarding experimental results of MANAS vs MANAS-LS:
> While MANAS-LS uses a strong assumption (linear decomposition of the loss), it also enables more direct coordination between different agents. MANAS does indeed make fewer assumptions but has less context-awareness. In our experiments none of the approaches dominates the other, implying a data-dependency. For example, on the Sport-8, Caltech-101 and MIT-67 datasets (except for the 14-cell model), MANAS does indeed outperform MANAS-LS.
>
> Regarding theoretical guarantees:
> Our theoretical analysis demonstrates that our algorithms are robust in the sense that we guarantee good performance under the worst-case scenario of an adversarial loss sequence. Concretely, a comparison between theoretical and practical results requires knowledge of the optimal solution to the problem in order to compute the regret. Since the optimal solution is intractable in the NAS problem, evaluating the regret in practice is not feasible. However, that analysis can be done in a smaller/tractable problem, and we analysed such a problem in the Gaussian Squeeze Domain experiment (GSD; Appendix F). That experiment confirms that, in practice, the empirical cumulative regret is indeed bounded by the theretical value, O(N*sqrt(T*K*log(K))). To make this comparison explicit, we have included a new figure in Appendix F (Figure 4) showing the comparison between the theoretical bound for the cumulative regret and the observed counterpart.
>
> Thank you for bringing to our attention the missing value in Table 1. This has now been added.

---

### Author Response · Authors · 2019-11-11
**Reply to all Reviewers**

We thank the reviewers for the helpful comments and will be happy to elaborate further on any of the issues raised.

To summarize, there seems to be consensus that a) the method is theoretically grounded and well motivated; b) the extensive experimental evaluation on 5 datasets is pioneering and highlights the robustness of the proposed approach.

While one reviewer has expressed concerns regarding the relatively small increase in accuracy on CIFAR10, we would like to stress that our contribution is mainly intended to deliver a computationally efficient (10 times less memory usage than gradient-based methods) and theoretically grounded alternative to existing solutions. Furthermore, as current results on CIFAR10 are already very high, we decided to also report results on three novel datasets (where we also outperform DARTS) to highlight the robustness of our approach.
Additionally, it is not a minor point that we are able to search for the optimal network architecture directly on large datasets (ImageNet), which many competing methods can't. The fact that the transfer results typically reported are relatively good relies heavily on the availability of a strong transfer dataset (CIFAR10), which is not the general case, especially in real-world applications.

Thanks to reviewers' feedback, we have clarified the key issues in the paper. To address the concerns raised, we have also added some additional results to support our claims.

---

### Decision · Program_Chairs · 2019-12-19

**Decision:**

Reject

**Comment:**

This paper introduces a NAS algorithm based on multi-agent optimization, treating each architecture choice as a bandit and using an adversarial bandit framework to address the non-stationarity of the system that results from the other bandits running in parallel.

Two reviewers ranked the paper as a weak accept and one ranked it as a weak reject. The rebuttal answered some questions, and based on this the reviewers kept their ratings. The discussion between reviewers and AC did not result in a consensus. The average score was below the acceptance threshold, but since it was close I read the paper in detail myself before deciding.

Here is my personal assessment:

"
Positives:
1. It is very nice to see some theory for NAS, as there isn't really any so far. The theory for MANAS itself does not appear to be very compelling, since it assumes that all but one bandit is fixed, i.e., that the problem is stationary, which it clearly isn't. But if I understand correctly, MANAS-LS does not have that problem. (It would be good if the authors could make these points more explicit in future versions.)

2. The absolute numbers for the experimental results on CIFAR-10 are strong.

3. I welcome the experiments on 3 additional datasets.

Negatives:
1. The paper crucially omits a comparison to random search with weight sharing (RandomNAS-WS) as introduced by Li & Talwalkar's paper "Random Search and Reproducibility for Neural Architecture Search" (https://arxiv.org/abs/1902.07638), on arXiv since February and published at UAI 2019. This method is basically MANAS without the update step, using a uniform random distribution at step 3 of the algorithm, and therefore would be the right baseline to see whether the bandits are actually learning anything. RandomNAS-WS has the same memory improvements over DARTS as MANAS, so this part is not new. Similarly, there is GDAS as another recent approach with the same low memory requirement: http://openaccess.thecvf.com/content_CVPR_2019/html/Dong_Searching_for_a_Robust_Neural_Architecture_in_Four_GPU_Hours_CVPR_2019_paper.html
This is my most important criticism.

2. I think there may be a typo somewhere concerning the runtimes of MANAS. It would be extremely surprising if MANAS truly takes 2.5 times longer when run with 20 cells and 500 epochs than when run with 8 cells and 50 epochs. It would make sense if MANAS gets 2.5 slower when just going from 8 to 20 cells, but when going from 50 to 500 epochs the cost should go up by another factor of 10. And the text states specifically that "for datasets other than ImageNet, we use 500 epochs during the search phase for architectures with 20 cells, 400 epochs for 14 cells, and 50 epochs for 8 cells". Therefore, I think either that text is wrong or MANAS got 10x more budget than DARTS.

3. Figure 2 shows that on Sport-8, MANAS actually does *significantly worse* when searching on 14 cells than on 8 cells (note the different scale of the y axis). It's also slightly better with 8 cells on MIT-67. I recommend that the authors discuss this in the text and offer some explanation, rather than have the text claim that 14 cells are better and the figure contradict this. Only for MANAS-LS, the 14-cell version actually works better.

4. The authors are unclear about whether they compare to random search or random sampling. These are two different approaches. Random sampling (as proposed by Sciuto et al, 2019) takes a single random architecture from the search space and compares to that. Standard random search iteratively samples N random architectures and evaluates them (usually on some proxy metric), selecting and retraining the best one found that way. The number N is chosen for random search to use the same computational resources as the method being compared. The authors call their method random search but then appear to be describing random sampling.

Also, with several recent papers showcasing problems in NAS evaluation (many design decisions affect NAS performance), it would be a big plus to have code available to ensure reproducibility. Many ICLR papers are submitted with an anonymized code repository, and if possible, I would encourage the authors to do this for a future version.
"

The prior rating based on the reviewers was slightly below the acceptance threshold, and my personal judgement did not push the paper above the acceptance threshold. I encourage the authors to improve the paper by addressing the reviewer's points and the points above and resubmit to a future venue. Overall, I believe this is very interesting work and am looking forward to a future version.